# Compression Represents Intelligence Linearly

**Yuzhen Huang**[*1]  **Jinghan Zhang**[*1]  **Zifei Shan**[2]  **Junxian He**[1]
[1]The Hong Kong University of Science and Technology  [2]Tencent
{yhuanghj,jzhangjv,junxianh}@cse.ust.hk

## Abstract

There is a belief that learning to compress well will lead to intelligence (Hutter, 2006). Recently, language modeling has been shown to be equivalent to compression, which offers a compelling rationale for the success of large language models (LLMs): the development of more advanced language models is essentially enhancing compression which facilitates intelligence. Despite such appealing discussions, little empirical evidence is present for the interplay between compression and intelligence. In this work, we examine their relationship in the context of LLMs, treating LLMs as data compressors. Given the abstract concept of "intelligence", we adopt the average downstream benchmark scores as a surrogate, specifically targeting intelligence related to knowledge and commonsense, coding, and mathematical reasoning. Across 12 benchmarks, our study brings together 31 public LLMs that originate from diverse organizations. Remarkably, we find that LLMs' intelligence – reflected by average benchmark scores – almost *linearly* correlates with their ability to compress external text corpora. These results provide concrete evidence supporting the belief that superior compression indicates greater intelligence. Furthermore, our findings suggest that compression efficiency, as an unsupervised metric derived from raw text corpora, serves as a reliable evaluation measure that is linearly associated with the model capabilities. We open-source our compression datasets as well as our data collection pipelines to facilitate future researchers to assess compression properly.[1]

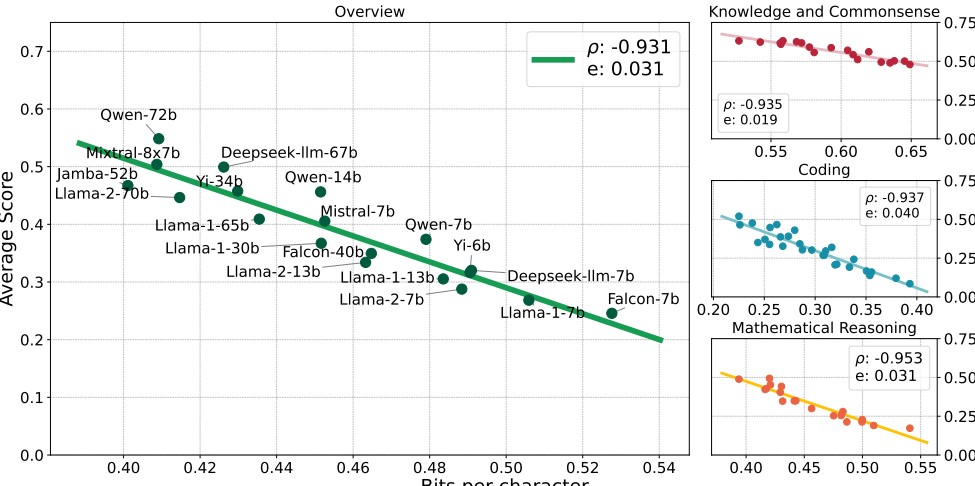

Figure 1: Correlation between the average benchmark scores and the models' compression efficiency evaluated with bits per character (BPC). BPC is computed on external raw corpora as listed in Table 1. Each point represents an LLM. **Left** shows the average benchmark score v.s. average BPC across, while **Right** focuses on three specific areas. $\rho$ and $e$ denote the Pearson correlation coefficient and the root mean square error of the linear fit respectively.

---

[*]Equal Contribution.
[1]https://github.com/hkust-nlp/llm-compression-intelligence.

# 1 Introduction

The belief that compression is closely connected to intelligence has been held for a long time, with some researchers suggesting that they are fundamentally equivalent (Hernández-Orallo & Minaya-Collado, 1998; Mahoney, 1999; Legg et al., 2005; Hutter, 2006; Legg & Hutter, 2007). This notion becomes even more compelling in light of the recent advancements in large language models (LLMs) and their implications for AI, leading to an exploration of language modeling from a compression standpoint. According to compression theory, any predictive model can be transformed into a lossless compressor and vice versa. Thus, language modeling can be considered a form of compression, with LLMs showing strong capabilities in data compression empirically (Deletang et al., 2024).

From this viewpoint, the argument that compression leads to intelligence becomes more relevant to the current paradigm of AI based on LLMs. Yet, despite these theoretical discussions, empirical evidence on the relationship between compression and intelligence remains limited. In this work, we seek to conduct such empirical study and answer: *If a language model can encode a text corpus with fewer bits in a lossless manner, does that indicate greater intelligence?* While the precise definition of "intelligence" often ventures into philosophical territory and can vary significantly, our work approaches "intelligence" from a practical standpoint, focusing on the model's ability to complete various downstream tasks. This treatment aligns with the definition from Hutter (2006), who notes that "intelligence is an agent's ability to achieve goals in a wide range of environments".

We measure intelligence along three key abilities: knowledge and commonsense, coding, and mathematical reasoning. Specifically, we begin by collecting external raw corpora in the respective domain (e.g. GitHub code for the coding ability) and evaluate the efficiency of various LLMs in compressing the corpus. Subsequently, we assess these models on a range of downstream tasks, using the average benchmark scores to gauge their domain-specific intelligence. The goal is to examine the correlation between the models' downstream ability and their compression efficiency. On experiments across 31 public LLMs and 12 diverse benchmarks, we uncover a remarkable finding: LLMs' downstream ability is almost *linearly* correlated with their compression efficiency, with a Pearson correlation coefficient of around -0.95 for each evaluated domain of intelligence as shown in Figure 1. Notably, this linear relationship even extends to most individual benchmarks as well.

Recent and concurrent works have explored the relationship between benchmark scores and compression-equivalent metrics like validation loss within the same model series, where the model checkpoints share most configurations such as model designs, tokenizers, and data (Xia et al., 2023; Wei et al., 2023; Gadre et al., 2024; Du et al., 2024). Our study, however, is the first to document a linear correlation between compression and intelligence in LLMs across varying model sizes, tokenizers, context window lengths, and pretraining data distributions. Our findings establish the linear correlation between compression and intelligence as a universal principle, providing empirical support for the longstanding belief that superior compression is indicative of greater intelligence.

From a practical perspective, compression efficiency serves as an unsupervised metric for LLMs where the text corpora can be easily updated to avoid overfitting or test contamination. Our findings advocate for adopting compression efficiency as a stable, flexible, and reliable metric to evaluate LLMs, which is linearly correlated with the models' abilities. We open-source our compression corpora, as well as the data collection and processing pipelines so that future researchers can easily collect and update their own corpora to assess compression.

# 2 Background: Language Modeling is Compression

The discussion on the equivalence between language modeling and compression has a long history (Shannon, 1948; 1951). Recently, Deletang et al. (2024) have echoed this connection and demonstrated that language models are general-purpose compressors empirically. Below, we briefly introduce the background to acquaint readers with such a relationship.

**Lossless Compression of Language:** Suppose $p_{data}$ is the underlying distribution to generate the text sequence data. Given a text sequence $x_{1:n}$, lossless compression seeks to encode it into a code $C(x_{1:n})$ that is smaller in size while preserving all original information. We typically use binary bits as the coding alphabet of $C$. According to the source coding theorem (Shannon, 1948), the expected number of bits of an optimal encoder is equal to $\mathbb{E}_{x \sim p_{data}}[-\log_2 p_{data}]$, which is the minimum average number of bits that one can achieve on compressing text from $p_{data}$ in a lossless manner. Knowing $p_{data}$, a common approach to achieve such optimal compression is arithmetic coding (Rissanen, 1976; Pasco, 1977), which is a practical approach to optimally encode data with a probabilistic model. We refer the readers to Deletang et al. (2024) for more details on arithmetic coding for autoregressive language models.

**Connection to Language Models:** In practice, however, $p_{data}$ is often unknown, and we can use a language model $p_{model}(x)$ to help compress $x$ efficiently. Intuitively, we seek to utilize $p_{model}(x)$ to encode rare sequences with more bits and the frequent ones with fewer bits. Arithmetic coding is able to achieve the optimal expected code length (in bits) as:

$$\text{Optimal \# Bits on Average} = \mathbb{E}_{x \sim p_{data}}[\sum_{i=1}^{n} -\log_2 p_{model}(x_i|x_{1:i-1})], \tag{1}$$

which establishes the connection between $p_{model}$ and compressors. To achieve more efficient compression, we aim to optimize $p_{model}$ to minimize the average code length in Eq. 1. Noting that Eq. 1 is exactly the cross-entropy loss that is used to train language models, learning to compress well is equivalent to language modeling.[2] This implies that if a language model has a lower loss on the data to be compressed, it will be able to compress it with fewer bits in a lossless manner. Strictly speaking, one needs to access the language model parameters to recover the original data from the code, thus the bits required to encode these parameters should be counted as well. However, such a cost can be amortized and becomes negligible when the model is used to compress a substantially large volume of data. There are online or offline settings to compress a data stream. In the online setting, the model is randomly initialized and trained on the data to be compressed. In the offline setting, the model is trained on external data and adopted to compress a different dataset. We focus on the offline setting in this paper, using pretrained language models to compress external corpora.

## 3 Examining Compression and Intelligence – the Experimental Protocols

Our goal is to empirically examine the relationship between compression and intelligence in the context of large language models. Our method is to investigate a variety of pretrained LLMs of different sizes and from diverse organizations, assessing their compression efficiency and "intelligence" respectively. Then we aim to summarize the potential relationship between compression and intelligence based on the evaluation results. Our overall principle is to collect diverse LLMs that are created with different training data, tokenizers, computation, and even architectures (e.g., mixture of experts as in MistralAI (2023)), so that our conclusion on compression and intelligence is general and agnostic to specific model series. Next, we detail our experimental protocols to evaluate compression and intelligence.

### 3.1 Evaluating Intelligence of LLMs

The definition of "intelligence" has been a subject of debate for years. Alan Turing firstly proposed the definition based on the Turing Test, which evaluates a machine's ability to exhibit intelligent behavior indistinguishable from that of a human (Turing, 1950). Although the Turing test is probably the most commonly accepted definition for intelligence, the application of the Turing Test as a measure for AI is deemed impractical, primarily because the evaluation's outcome relies heavily on the subjective judgment of the interrogator. More recently, Legg & Hutter (2007) proposed a definition termed *universal intelligence*, a broader interpretation than Turing's focus on human-like intelligence. They posited

---

[2]Libraries like PyTorch often implement the cross entropy loss with the log base of $e$ rather than 2, which causes a constant coefficient difference. Yet, it does not influence the equivalence.

| Ability | Source | # Character | Time Period | Benchmark |
|---|---|---|---|---|
| Knowledge and commonsense | Common Crawl | 131M | 23/09-23/10 | NQ (Kwiatkowski et al., 2019), TQA (Joshi et al., 2017), ARC-C (Clark et al., 2018), HellaSwag (Zellers et al., 2019), MMLU (Hendrycks et al., 2021a) |
| Coding | GitHub | 98M | 23/06-23/09 | HumanEval (Chen et al., 2021), MBPP (Austin et al., 2021), DS-1000 (Lai et al., 2023) |
| Mathematical reasoning | ArXiv | 101M | 23/09-23/10 | GSM8K (Cobbe et al., 2021), MATH (Hendrycks et al., 2021b), SAT-Math (Zhong et al., 2023), MMLU-Math |

Table 1: Overview of the compression corpora and benchmarks. "Time Period" refers to the start and end dates of the collected data – we use the latest data at the time of the experiments to mitigate data leakage. NQ, TQA, and ARC-C denote NaturalQuestions, TriviaQA, and ARC-Challenge respectively.

that an agent's capacity to achieve goals across a wide array of scenarios should define its intelligence. This practical approach to defining intelligence guides our evaluation of models, wherein we assess their intelligence based on average performance across multiple downstream tasks. Such practical evaluation is commonly adopted by most LLM developers to showcase the potential of their models (Touvron et al., 2023a;b; Jiang et al., 2023; Team et al., 2023), and Anthropic (2024) directly noted the "intelligence" of Claude 3 with the average benchmark scores. Specifically, in this paper, we study intelligence along three key abilities: knowledge and commonsense, coding, and mathematical reasoning. These abilities represent the most focused LLM areas nowadays, and we collect the well-recognized benchmarks for each of these areas, utilizing their average score to indicate domain-specific intelligence. The benchmarks are listed in Table 1.

### 3.2 Evaluating Compression of LLMs

**Metric:** According to §2, we evaluate the model's compression efficiency using the average coding length per message of compressing a given corpus. Due to the variety of tokenizers employed by different LLMs, the average bits per token are not directly comparable. Therefore, we utilize the average bits per character (BPC) as the metric. BPC is a commonly used compression metric for compressing text, such as in enwik8 (Mahoney, 2011). Specifically,

$$BPC = \frac{-\log_2 p_{model}(X)}{T} = \frac{\sum_{i=1}^{N} -\log_2 p_{model}(x_i|x_{1:i-1})}{T}, \quad (2)$$

where $X$ is the corpus to be compressed, $N$ is the total number of tokens of $X$ tokenized by the model's tokenizer, and $T$ is the total number of characters of $X$. Note that Eq. 2 is equal to the per-character language model loss on the corpus $X$ (with a constant log base shift).

**Context Window Unification:** LLMs can have different context window sizes, and a longer context window gives advantages to compression efficiency. This is because a larger context window offers more information to predict the next token's distribution and allows for encoding with fewer bits, as illustrated in Eq. 2. However, in downstream benchmarks where input lengths are short, the benefit of extended context windows diminishes. This applies to all our benchmarks in Table 1, where the inputs do not exceed 2048 tokens, even in few-shot in-context learning scenarios. Basically, the downstream tasks in our study only reflect the model's intelligence when operating within relatively short contexts. To study its correlation with compression, it is crucial to assess compression using comparable context length, ensuring consistency in the models' access to information during both corpus compression and execution of downstream tasks. Therefore, for all our compression and benchmark evaluations across different LLMs, we unify the context window size to be 1900 tokens that are sufficient for all benchmark evaluation. Ideally, a more holistic evaluation of intelligence should incorporate tasks involving both short and long contexts. However, in this paper, our focus is on the short- or medium-context regime, which encompasses the majority of benchmark tasks. We leave the exploration of long-context scenarios for future work. Apart from context window unification, we utilize a sliding window approach to evaluate compression more accurately, as detailed in Appendix A.1.

**Focusing on Base Models:** In the development of LLMs, there are typically two stages: the pre-training stage and the alignment stage. The models are referred to as base models after the pre-training stage and as fine-tuned models following the alignment stage. We note that fine-tuned models are no longer general-purpose compressors since it does not model the next token distribution for arbitrary text, but only for structured (query, response) data. Besides, it is commonly believed that the intelligence of LLMs is learned during the pretraining stage and remains relatively fixed during the alignment stage (Zhou et al., 2023). Therefore, in this work, we focus on the base models only, while in §5 we will further discuss the case of compression v.s. intelligence for fine-tuned models.

**Compression Corpus:** What kind of corpus shall we select to measure the models' compression? Firstly, it is important to recognize that different corpora can illuminate various aspects of models' abilities, and compressing one corpus well may not generalize to another (Magnusson et al., 2023). Therefore, we would like to select the corpora that align with the areas of our focus. Secondly, the chosen corpora should not intersect with the models' pretraining data to avoid data leakage. Given the opaque status of LLMs' pre-training datasets, we opt to use the newest corpora as a measure. Concretely, for assessing knowledge and commonsense, we have compiled texts from the latest Common Crawl dataset. To evaluate coding ability, we have sourced data from GitHub repositories mainly on the Python language since the downstream benchmarks focus on Python coding abilities. For mathematical reasoning, we collect academic papers from ArXiv, specifically selecting those designated with "math" as their primary category. For each data source, we ensure the recency by utilizing the latest available data at the time of our experiments. The composition of our compression corpora is detailed in Table 1. More details about data collection and processing are in Appendix A.2.

### 3.3 Overfitting Benchmarks

LLMs may be overoptimized towards certain benchmarks, in this case, the benchmark scores are not a good proxy for intelligence anymore.[3] For example, Bi et al. (2024) show that when adding multi-choice QA training data, the model is able to achieve over 10 absolute points improvement on multi-choice QA benchmarks such as MMLU, while its performance on general QA benchmarks like TriviaQA remains unchanged. In the math domain, recent researches find that some models heavily optimize towards the GSM8K and MATH benchmarks, while performing poorly on held-out math tests (Paster, 2023). In such scenarios, these benchmark scores are not reliable enough to represent the models' intelligence in the respective area. Although we focus on base models in this work, where such overfitting phenomenon may be less common, it is still possible that some LLMs are specially trained on the training data of the corresponding benchmarks in the pretraining stage, or even worse, suffer from test data contamination issues as evidenced in Wei et al. (2023). To identify such cases, we adopt the MIN-K% PROB method (Shi et al., 2024) which is proposed to detect whether the given text is exposed to the model during pretraining. The MIN-K% PROB approach selects the $k\%$ tokens in a given example with minimum probabilities, if the average probability of these tokens is high, then the example is likely to be present in the pretraining data. In this approach, the MIN-K% Score is defined to be the average negative log-likelihood of these selected tokens. In the experiments next, we compute the MIN-K% Score for all the test splits of the benchmarks, and for the training splits as well if available. We will spot and discuss the LLMs with extremely abnormal MIN-K% Scores. We choose $k$ to be 20 in our experiments as suggested by Shi et al. (2024).

## 4 How is Intelligence Correlated with Compression?

**General Setup:** To provide a comprehensive study of the correlation between compression efficiency and intelligence, we include 9 series of general-purpose language models, covering diverse organizations, varying in size and architectures, as shown in Table 2. Notably, in addition to dense transformer models, we further include Mixtral-8x7B (MistralAI, 2023)

---

[3]"When a measure becomes a target, it ceases to be a good measure" – Goodhart's law.

| Series | Creator | # Parameters |
|---|---|---|
| *General-purpose Language Model* | | |
| Deepseek-llm (Bi et al., 2024) | DeepSeek | 7B, 67B |
| Falcon (Almazrouei et al., 2023) | TII | 7B, 40B |
| Jamba (Lieber et al., 2024) | AI21 | 52B |
| Llama-1 (Touvron et al., 2023a) | Meta | 7B, 13B, 30B, 65B |
| Llama-2 (Touvron et al., 2023b) | Meta | 7B, 13B, 70B |
| Mistral (Jiang et al., 2023) | MistralAI | 7B |
| Mixtral-8x7B (MistralAI, 2023) | MistralAI | 8x7B |
| Qwen (Bai et al., 2023) | Alibaba | 7B, 14B, 72B |
| Yi (AI et al., 2024) | 01.AI | 6B, 34B |
| *Code Language Model* | | |
| CodeLlama (Roziere et al., 2023) | Meta | 7B, 13B, 34B |
| Deepseek-coder (Guo et al., 2024) | DeepSeek | 1.3B, 6.7B, 33B |
| StarCoderBase (Li et al., 2023) | BigCode | 15.5B |
| StarCoder2 (Lozhkov et al., 2024) | BigCode | 7B, 15B |
| *Mathematical Language Model* | | |
| Deepseek-math (Shao et al., 2024) | DeepSeek | 7B |
| Llemma (Azerbayev et al., 2023) | EleutherAI | 7B, 34B |

Table 2: Models evaluated in this paper.

and Jamba-52B (Lieber et al., 2024) to validate our findings across different architectures. Mixtral-8x7B utilizes a sparse mixture-of-experts (MoE) architecture, whereas Jamba-52B is based on a hybrid Transformer-Mamba MoE architecture. Furthermore, we incorporate state-of-the-art models specialized in coding and mathematical reasoning in addition to general-purpose models, as outlined in Table 2. We assess general-purpose LLMs across all benchmarks, while we include code LLMs in the code area only and mathematical LLMs in the mathematical reasoning area only. We note that our investigation focuses exclusively on well-trained LLMs, implying that the intelligence evaluated is likely already manifest, in contrast to models where such abilities have yet to emerge (Wei et al., 2022). This is because LLMs that are not optimized enough tend to perform poorly, where the results are overly noisy and less meaningful, complicating the analysis. Models are evaluated using few-shot in-context learning or in a zero-shot manner, adhering to the established norms for the respective benchmarks. All the models are evaluated exactly the same way with the same codebase for a fair comparison. More details on evaluation are included in Appendix A.3.

**Metrics:** As described in §3.2, we utilize bits per character (BPC) as the evaluation metric for compression efficiency. To quantitatively assess the correlation between intelligence and compression, we report the Pearson correlation coefficient (Pearson $\rho$) between the average benchmark scores and BPC. Based on preliminary observations that the correlation is highly linear, we perform a linear fit between the average benchmark scores and BPC, and report the Root Mean Square Error (RMSE $e$) that captures how much the models' benchmark scores deviate from the linear prediction on average. We extend this analysis to individual benchmarks as well in addition to the average. Qualitatively, we plot the linear regression results to visualize the relationship, with each model being a point in the figure.

### 4.1 Main Results – Compression Represents Intelligence Linearly

**Overall intelligence v.s. compression:** We have summarized our main results in Figure 1, where we examine the overall relationship by computing the average benchmark scores across all three areas, and the average BPC across all the compression corpora. The average score and the average BPC demonstrate a highly linear correlation qualitatively in the visualization. Quantitative results with a Pearson correlation coefficient of -0.93 and an RMSE of 3.1% further verify the linear relationship. We consider such a linear correlation significant given the noisy nature of evaluating on downstream benchmarks – it is known that varying the prompts or decoding hyperparameters could easily cause several points of difference for the task scores (Lu et al., 2022; Gonen et al., 2022; Sclar et al., 2024). We also report the results for the three key areas respectively which demonstrate similar phenomena qualitatively and quantitatively. We note that several previous and concurrent works have studied the correlation between benchmark scores and validation losses in limited settings –

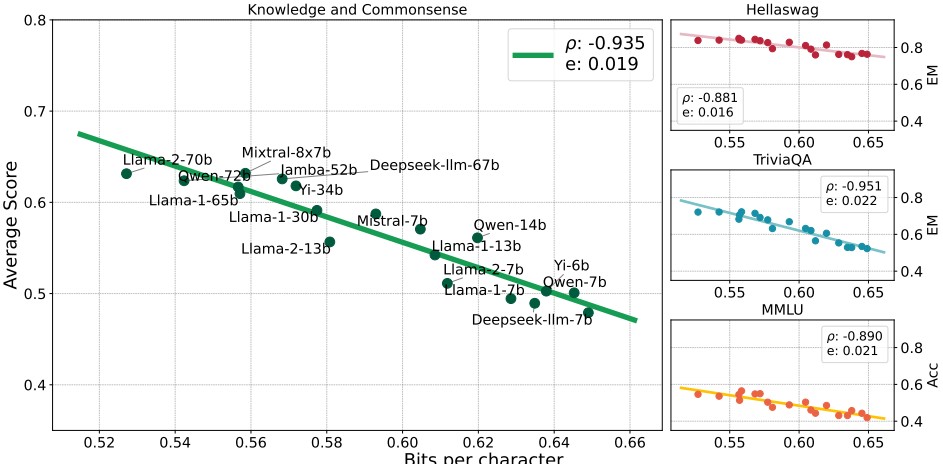

Figure 2: **Left:** Correlation between benchmark scores and BPC in the knowledge and commonsense area. **Right:** The correlation on Hellaswag, TriviaQA and MMLU respectively. Results on NQ and ARC-C are shown in Figure 9(a) and Figure 9(b). BPC is computed on a collection of Common Crawl.

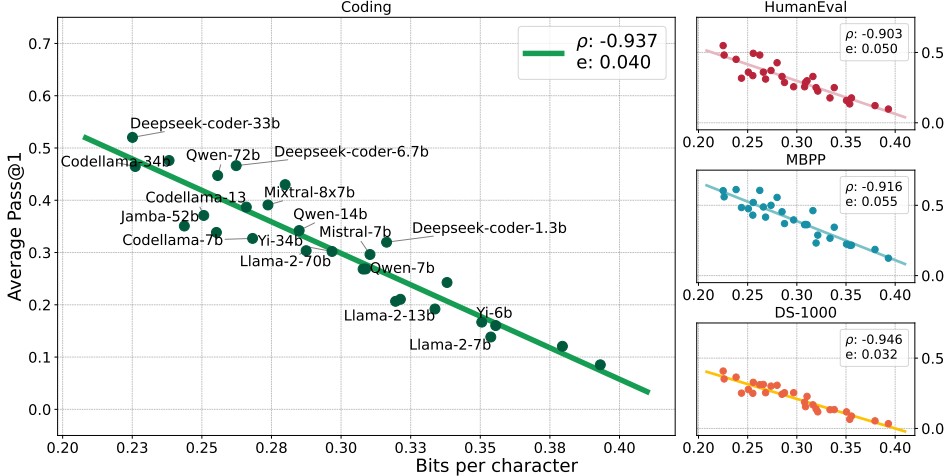

Figure 3: **Left:** Correlation between average benchmark scores and BPC in the coding area. **Right:** The correlation on HumanEval, MBPP, and DS-1000. BPC is computed on a collection of GitHub repos.

they focus on either the model checkpoints of the same model over the course of pretraining or the same model series (Xia et al., 2023; Lu et al., 2023; Wei et al., 2023; Gadre et al., 2024; Du et al., 2024). Their studied checkpoints share designs and tokenizers and are pretrained on the same data distributions. However, our work escapes the constraints on the same model (series) and compares across models that diverge dramatically on tokenizers, model designs, and pretraining data. Our results mark the linear correlation between compression and intelligence in a general manner, establishing it as a universal principle. The specific benchmark scores and BPC for every model are reported in Appendix B. Next, we discuss the results in the three areas with more details respectively, and extend the examination to individual benchmarks as well.

**Knowledge and Commonsense:** We present both the average and individual benchmark results for the knowledge and commonsense area in Figure 2. Notably, the scores from individual benchmarks exhibit a strong linear correlation with compression efficiency, indicated by Pearson coefficients of -0.881, -0.951, and -0.890 for HellaSwag, TriviaQA, and MMLU, respectively. The Root Mean Square Error $e$ is approximately 2 absolute points, which falls within the expected range of variation – Minor fluctuations in evaluation details, such as prompt adjustments, can readily account for a 2-point accuracy shift (Sclar et al.,

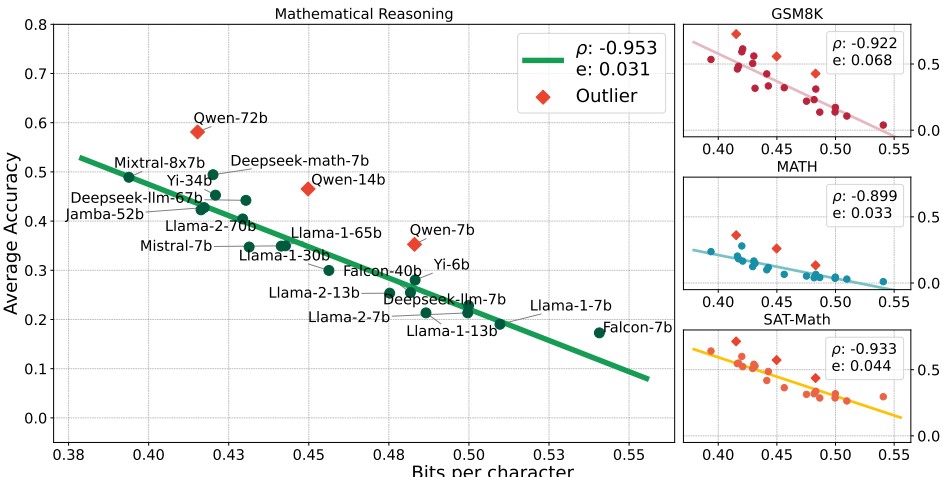

Figure 4: **Left:** Correlation between average benchmark scores and BPC in the mathematical reasoning area. **Right:** The correlation on GSM8K, MATH, and SAT-Math. Results on MMLU-Math is in Figure 9(c). BPC is computed on a collection of ArXiv papers with "math" tag. Outliers denote the models are likely overfitting the benchmarks, which are detected by the Min-K% PROB method as detailed in §3.3.

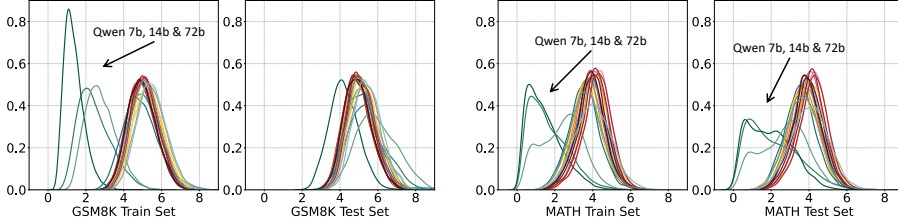

Figure 5: Distribution of MIN-K% scores calculated with assessed models across four datasets, displayed in a linear array ($k = 20$). The y-axis represents the density of distributions, where higher values indicate a greater concentration of scores, achieved through kernel density estimation.

2024). Meanwhile, we observe that the linear correlation in both HellaSwag and MMLU is not as pronounced as in TriviaQA, albeit for different reasons. For HellaSwag, the average accuracy among models has exceeded 0.8, hinting at a saturation point in performance that blurs the distinctions between models. Conversely, MMLU, which draws heavily from real-world examinations and textbooks, presents a slight mismatch in distribution compared to the Common Crawl dataset used to measure compression efficiency. Further experiments on MMLU and textbook compression are detailed in Appendix E, where a Pearson coefficient of -0.930 is observed. We note that for MMLU evaluation, we employ a cloze-style input format for this popular multiple-choice question (MCQ) benchmark, presenting only the question without the options, and subsequently selecting the choice with the highest likelihood. We intentionally opted for this less conventional evaluation style to minimize the risk of models overfitting to the MCQ format, thus over-optimizing the scores. Please see Appendix D for detailed discussion and ablation.

**Coding:** Similar to the knowledge and commonsense area, there is a strong linear correlation between the models' compression efficiency and its coding ability, with the Pearson correlation coefficient within $[-1, -0.9]$ for each individual benchmark. Furthermore, despite having significantly different architectures, both Mixtral-8x7B and Jamba-52B fit well into this linear correlation, demonstrating that our findings are agnostic to model architecture. We notice the strongest linear correlation within the DS-1000 benchmark, with a Pearson coefficient of -0.946, surpassing those of HumanEval and MBPP. We think that this is because we collect Python code from the popular GitHub repositories in recent months, which are mostly related to data science and machine learning, aligned closer with DS-1000.

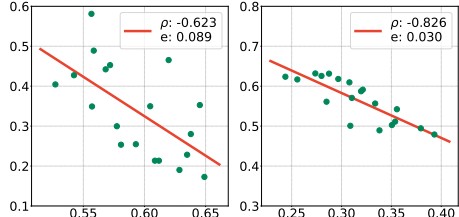
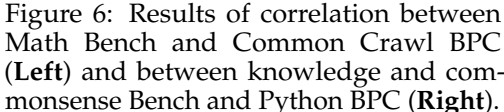
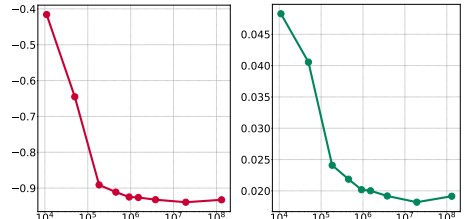

Figure 6: Results of correlation between Math Bench and Common Crawl BPC (**Left**) and between knowledge and commonsense Bench and Python BPC (**Right**).

Figure 7: Variation of the Pearson coefficient (**Left**) and the RMSE (**Right**) with respect to #characters of the CC corpus in knowledge and commonsense ability.

We note that the Deepseek-coder model series are consistently well-above the linear fit, demonstrating strong abilities on the HumanEval and MBPP benchmarks. We hypothesize that this is due the fact that Deepseek-coder models are exposed to private, task-specific data during pretraining that help HumanEval and MBPP significantly, yet we do not have reliable methods to detect the existance of such data and verify this hypotheis.

**Mathematical Reasoning:** Overall the correlation results in the mathematical reasoning area are similar to the other two, as shown in Figure 4. However, overfitting benchmarks in mathematical reasoning warrants particular attention. As described in §3.3, we compute the MIN-K% Score for every example – which is the average negative log likelihood of the $k$% tokens with the lowest probabilities – on the benchmark test data as well as training data if available. Abnormally low MIN-K% Scores imply that the example is likely to be present in the model's pretraining data. We did not observe abnormal MIN-K% patterns in all the previous experiments, yet, there are models with far lower Min-K% Scores on the GSM8K and MATH datasets than all other models. We visualize the MIN-K% distribution of the training and test examples on both GSM8K and MATH in Figure 5, which implies that the Qwen model series may be exposed to the GSM8K training data, MATH training data, and even the MATH test data in the pretraining stage. Therefore, we spot the Qwen model series in our visualization and exclude them when performing the linear fit and compute the correlation metrics. As expected, the three Qwen models achieve generally higher accuracies than the predicted performance from the linear fit, and appear as outliers in Figure 4.

At last, we highlight that the strongly linear correlation between the single benchmark scores and compression efficiency was originally unexpected, because individual benchmark scores are typically too noisy to represent models' general ability. However, current findings indicate that these individual scores may be predictable from the BPC with a minor error in most cases indicated by small RMSE.

## 4.2 Remaining Questions

**How does different compression corpus influence the linear relationship?** In the previous experiments, we intentionally selected the appropriate compression corpora aligning with the area of focus. Herein, we further explore the effects of diverse compression corpora on the results. We investigate two representative examples where the compression corpus does not align with the specific domain, to study whether an arbitrary compression corpus would suffice for a strong linear correlation: (1) mathematical reasoning benchmarks v.s. compression on Common Crawl data, and (2) knowledge and commonsense benchmarks v.s. compression on GitHub code data. We present the findings in Figure 6. Due to a substantial ability mismatch between the compression corpus and the benchmarks, the linear correlation is significantly weaker than that with in-domain data. Coupled with the MMLU case discussed in Appendix E, these instances demonstrate that linear correlation relies on the alignment degree between the compression corpus and the specific area of study. However, we emphasize that our previous findings are relatively robust since our chosen compression corpora as listed in Table 1 rely on simple heuristics at a coarse granularity without meticulous designing. How to automatically identify the most appropriate

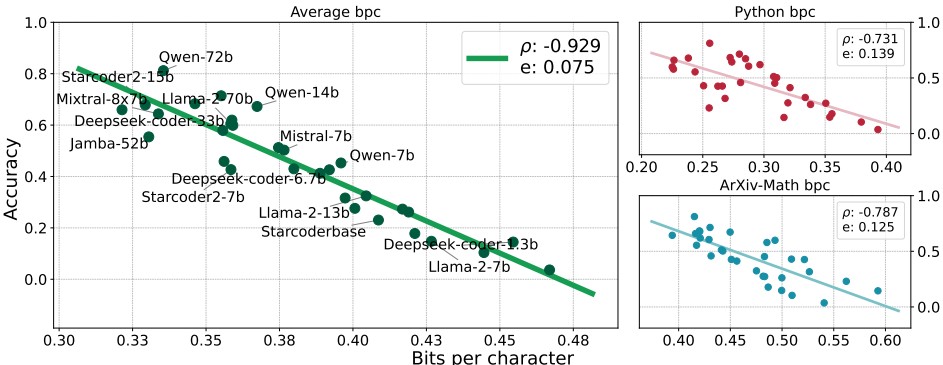

Figure 8: **Left:** Correlation between PAL-GSM8K scores and BPC on the combined Python and ArXiv-Math corpus. **Right:** The correlation between PAL-GSM8K scores and BPC metrics within Python and ArXiv-Math corpus, respectively.

compression corpus that correlates with certain benchmarks is an important problem that we leave for future work.

**How many characters are required to compute BPC reliably?** As an example, we investigate the effect of size of compression corpus on the knowledge and commonsense ability. We vary the number of characters of the compression corpus from 50K to 100M, observing changes in the Pearson correlation coefficient and the RMSE depicted in Figure 7. As the number of sampled characters increases, the linear correlation gradually strengthens and reaches saturation at around 3M. This observation suggests that the compression corpus is sufficiently large with only tens of millions of characters. We further provide an analysis of the impact of random sampling on compression evaluation in Appendix C.

**What about extending to cross-ability tasks?** The previously discussed experiments focus exclusively on tasks related to a single ability. Nevertheless, numerous tasks require abilities in multiple aspects. For example, incorporating programming as an intermediate step for reasoning has been shown to enhance performance in mathematical tasks, requiring abilities in both coding and mathematical reasoning (Gao et al., 2023b; Chen et al., 2023). Following Gao et al. (2023b), we apply the Program-Aided Math Reasoning (PAL) method to GSM8K. To align with the abilities involved, we combined the Python and ArXiv-Math corpora to create a mixed corpus. As illustrated in Figure 8, this mixed corpus exhibits a stronger linear correlation than using either the Python or ArXiv-Math corpora alone.

## 5 Discussion

**Compression as a reliable evaluation metric:** Our findings provide evidence for the belief that superior compression is indicative of greater intelligence. Practically, our experiments strengthen the rationale to adopt compression efficiency as an unsupervised, flexible, and reliable metric to assess LLMs' abilities. The compression corpus can be easily updated and composed flexibly, which mitigates the data contamination and benchmark overfitting issues that most evaluations suffer from. While recent work establishes a large collection of diverse corpora to evaluate LLMs with losses (Magnusson et al., 2023), our work offers strong empirical support for doing so by demonstrating the correlation with intelligence.

**Limitations:** Our study admits several limitations. First, we only focus on base models because fine-tuned models are not general-purpose compressors for arbitrary text. However, we posit that there are still interesting relationships to be explored about the base model compression efficiency and the benchmark scores of the corresponding fine-tuned models. Yuan et al. (2023) initially provides evidence of this correlation in mathematical reasoning ability, albeit limited to LLaMA models. Second, our focus lies on the short- to medium-context regimes, deferring the examination of long-context scenarios. Third, our conclusion and findings may only apply to well-trained models and not hold for LMs where the evaluated abilities have not emerged yet. We leave study of these problems for future work.

## Acknowledgements

We thank Graham Neubig, Yao Fu, and Pengcheng Yin for their valuable feedbacks for this draft.

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

# A  Detailed Setup of the Main Experiment

## A.1  Sliding Window

We evaluate models' compression with a sliding window, characterized by the repetitive sliding of the context window at a fixed stride of 512. This method, widely adopted in language modeling evaluation (Baevski & Auli, 2018; Press et al., 2021; Borgeaud et al., 2022), provides more contextual information. It effectively reduces the initial spike in metric observed at the start of the input segment. Furthermore, this approach better aligns with common application scenarios, which rely on preceding context for predictions.

## A.2  Data Collection

**Data Processing**    To mitigate data leakage, we collect the most recent data available at the time of the experiments. The data sources and corresponding collection periods are detailed in Table 1. In this study, we utilize standard data processing frameworks to collect and refine our validation data. For knowledge and commonsense ability, we download the Common Crawl dump number "2023-40", which comprises web data crawled between September 21st and October 5th. And then we leverage the CCNet pipeline (Wenzek et al., 2020) to streamline the data cleaning processes. For coding ability, we access repositories containing Python code via the GitHub GraphQL API[4], and apply the CodeParrot pipeline (CodeParrot, 2021) to filter out low-quality content. For mathematical reasoning ability, we obtain the September 2023 LaTeX source file dump via ArXiv's S3 Bulk Source File Access[5]. We then convert the LaTeX source files to Markdown using the pandoc library[6] and selectively retain those papers that list "math" as their primary tag. For more implementation details, please refer to our GitHub repository[7].

**Data Sampling**    Through the process mentioned above, we collect more data records than needed. Therefore, it becomes necessary to downsample the data. We employ a document-level random sampling strategy for our data, except Python code. Code files within the same repository frequently exhibit interdependency. To maintain a certain level of interdependency while ensuring the diversity of the repository, we implement a trade-off solution – segment-level sampling, which divides each repository into segments and then samples at segment level, thereby maintaining interdependency within segments.

## A.3  Benchmark Evaluation

For benchmark evaluation, all models are assessed following the common-used protocol, as detailed in Table 3. We utilize two types of evaluation methods: the perplexity-based method and the generation-based method. The perplexity-based method involves calculating and normalizing the perplexity for each option and selecting the option with the lowest perplexity. And for the generation-based method, we use greedy decoding uniformly. For the overall results presented in Figure 1 (**Left**), the average benchmark score (y-axis) is calculated by averaging the scores across 12 benchmarks, while the average bits per character (x-axis) is determined by averaging the BPC across three distinct areas.

# B  Detailed Results of Model Evaluation

**Knowledge and Commonsense Ability:**    The correlations on NaturalQuestions and ARC-Challenge are shown in Figure 9(a) and Figure 9(b), respectively. The detailed evaluation results of knowledge and commonsense ability are shown in Table 4. The breakdown of the

---

[4]https://docs.github.com/en/graphql

[5]https://info.arxiv.org/help/bulk_data_s3.html

[6]https://pandoc.org/

[7]https://github.com/hkust-nlp/llm-compression-intelligence

| Ability | Benchmark | N-shot | Evaluation Method | Codebase | Metric |
|---|---|---|---|---|---|
| Knowledge and commonsense | NaturalQuestions | 5 | Generation-based | LM-Evaluation-Harness | Exact match |
| | TriviaQA | 5 | Generation-based | LM-Evaluation-Harness | Exact match |
| | ARC-Challenge | 25 | Perplexity-based | LM-Evaluation-Harness | Acc_norm |
| | Hellaswag | 0 | Perplexity-based | LM-Evaluation-Harness | Acc_norm |
| | MMLU | 5 | Perplexity-based | LM-Evaluation-Harness | Acc_norm |
| Coding | HumanEval | 0 | Generation-based | Deepseek-Coder | Pass@1 |
| | MBPP | 3 | Generation-based | InCoder | Pass@1 |
| | DS-1000 | 0 | Generation-based | Deepseek-Coder | Pass@1 |
| Mathematical reasoning | GSM8K | 5 | Generation-based | LM-Evaluation-Harness | Acc |
| | MATH | 4 | Generation-based | OpenCompass | Acc |
| | SAT-Math | 4 | Generation-based | AGIEval | Acc |
| | MMLU-Math | 5 | Perplexity-based | LM-Evaluation-Harness | Acc_norm |

Table 3: Evaluation details of benchmarks. The perplexity-based evaluation method involves calculating and normalizing the perplexity for each option and selecting the option with the lowest perplexity. For the majority of benchmarks, we utilize the LM-Evaluation-Harness framework (Gao et al., 2023a). Additionally, we incorporate other sources as supplementary, including Deepseek-Coder (Guo et al., 2024), InCoder (Fried et al., 2023), OpenCompass (Contributors, 2023) and AGIEval (Zhong et al., 2023).

| Model | HellaSwag | ARC-Challenge | NaturalQuestions | TriviaQA | MMLU | Average | BPC on CC ($\downarrow$) |
|---|---|---|---|---|---|---|---|
| *6-7B Parameter Model* | | | | | | | |
| Deepseek-llm-7b | 76.1 | 52.5 | 20.1 | 52.9 | 43.1 | 48.9 | 0.635 |
| Falcon-7b | 76.3 | 47.5 | 21.4 | 52.3 | 41.9 | 47.9 | 0.649 |
| Llama-1-7b | 76.2 | 51.1 | 21.4 | 55.3 | 43.1 | 49.4 | 0.629 |
| Llama-2-7b | 76.0 | 54.1 | 24.7 | 56.5 | 44.3 | 51.1 | 0.612 |
| Mistral-7b | 81.1 | 61.2 | 29.5 | 63.2 | 50.3 | 57.1 | 0.605 |
| Qwen-7b | 76.8 | 52.2 | 23.7 | 53.4 | 44.3 | 50.1 | 0.645 |
| Yi-6b | 75.0 | 54.9 | 22.9 | 52.8 | 45.8 | 50.3 | 0.638 |
| *13-15B Paramter Model* | | | | | | | |
| Llama-1-13b | 79.1 | 56.1 | 27.8 | 62.1 | 46.1 | 54.2 | 0.609 |
| Llama-2-13b | 79.4 | 59.0 | 29.2 | 63.1 | 47.6 | 55.7 | 0.581 |
| Qwen-14b | 81.3 | 58.3 | 31.8 | 60.6 | 48.6 | 56.1 | 0.620 |
| *30-40B Parameter Model* | | | | | | | |
| Falcon-40b | 82.8 | 62.8 | 32.3 | 66.9 | 48.9 | 58.7 | 0.593 |
| Llama-1-30b | 82.6 | 61.9 | 32.8 | 67.9 | 50.3 | 59.1 | 0.577 |
| Yi-34b | 83.7 | 64.5 | 36.7 | 69.2 | 55.0 | 61.8 | 0.572 |
| *50-72B Parameter Model* | | | | | | | |
| Deepseek-llm-67b | 84.4 | 66.3 | 35.9 | 71.4 | 54.8 | 62.6 | 0.568 |
| Jamba-52b | 84.0 | 63.5 | 38.6 | 72.1 | 53.6 | 62.4 | 0.542 |
| Llama-1-65b | 84.2 | 63.2 | 35.5 | 70.6 | 51.4 | 61.0 | 0.557 |
| Llama-2-70b | 83.8 | 67.6 | 37.6 | 72.1 | 54.6 | 63.1 | 0.527 |
| Mixtral-8x7b | 84.0 | 66.0 | 37.1 | 72.2 | 56.5 | 63.2 | 0.559 |
| Qwen-72b | 85.0 | 64.0 | 36.7 | 68.2 | 54.5 | 61.7 | 0.557 |

Table 4: Results of knowledge and commonsense ability. We report models' compression efficiency on the Common Crawl (CC) dataset measured by bits per character.

| Benchmark | Hellaswag | ARC-Chanllange | NaturalQuestions | TriviaQA | MMLU | Average |
|---|---|---|---|---|---|---|
| Pearson $\rho$ | -0.881 | -0.913 | -0.925 | -0.951 | -0.890 | -0.935 |
| RMSE $e$ | 0.016 | 0.023 | 0.023 | 0.022 | 0.021 | 0.019 |

Table 5: The correlation for knowledge and commonsense ability. Here, Pearson $\rho$ denotes the Pearson correlation coefficient, and RMSE $e$ represents the root mean square error.

correlation between benchmark scores and compression efficiency on the validation data is presented in Table 5.

**Coding Ability:** The detailed evaluation results of coding ability are shown in Table 6. The breakdown of the correlation between benchmark scores and compression efficiency on the validation data is presented in Table 7.

| Model | HumanEval | Mbpp | DS-1000 | Average | BPC on Python Code (↓) |
|---|---|---|---|---|---|
| *1-7B Parameter Model* | | | | | |
| Codellama-7b | 31.1 | 41.6 | 25.4 | 32.7 | 0.268 |
| Deepseek-coder-1.3b | 32.9 | 46.2 | 16.8 | 32.0 | 0.316 |
| Deepseek-coder-6.7b | 48.2 | 60.6 | 31.1 | 46.6 | 0.262 |
| Deepseek-llm-7b | 25.0 | 34.4 | 13.4 | 24.3 | 0.338 |
| Falcon-7b | 9.8 | 12.4 | 3.4 | 8.5 | 0.393 |
| Llama-1-7b | 12.2 | 18.6 | 5.4 | 12.1 | 0.379 |
| Llama-2-7b | 13.4 | 21.6 | 6.5 | 13.8 | 0.354 |
| Mistral-7b | 29.9 | 36.2 | 22.8 | 29.6 | 0.310 |
| Qwen-7b | 28.7 | 36.4 | 15.5 | 26.9 | 0.309 |
| Starcoder2-7b | 36.0 | 48.8 | 31.3 | 38.7 | 0.266 |
| Yi-6b | 15.9 | 22.4 | 11.8 | 16.7 | 0.351 |
| *13-15B Paramter Model* | | | | | |
| Codellama-13 | 36.0 | 47.6 | 27.7 | 37.1 | 0.251 |
| Llama-1-13b | 17.7 | 21.6 | 8.8 | 16.0 | 0.356 |
| Llama-2-13b | 17.7 | 26.6 | 13.3 | 19.2 | 0.334 |
| Qwen-14b | 32.9 | 45.4 | 24.3 | 34.2 | 0.285 |
| Starcoder2-15b | 45.1 | 61.2 | 36.5 | 47.6 | 0.238 |
| StarcoderBase | 33.5 | 43.0 | 25.0 | 33.8 | 0.255 |
| *30-40B Parameter Model* | | | | | |
| Codellama-34b | 48.2 | 56.0 | 35.2 | 46.5 | 0.226 |
| Deepseek-coder-33b | 54.9 | 60.4 | 40.8 | 52.0 | 0.225 |
| Falcon-40b | 25.0 | 23.2 | 13.8 | 20.7 | 0.320 |
| Llama-1-30b | 22.6 | 28.8 | 11.8 | 21.1 | 0.321 |
| Yi-34b | 25.6 | 39.6 | 25.4 | 30.2 | 0.297 |
| *50-72B Parameter Model* | | | | | |
| Deepseek-llm-67b | 42.7 | 55.6 | 30.7 | 43.0 | 0.280 |
| Jamba-52b | 31.7 | 48.4 | 25.1 | 35.1 | 0.244 |
| Llama-1-65b | 25.6 | 36.2 | 18.7 | 26.8 | 0.308 |
| Llama-2-70b | 28.7 | 37.0 | 25.4 | 30.4 | 0.287 |
| Mixtral-8x7b | 37.2 | 50.0 | 30.1 | 39.1 | 0.274 |
| Qwen-72b | 49.4 | 52.0 | 32.8 | 44.7 | 0.256 |

Table 6: Results of coding ability. We report models' compression efficiency on the Python dataset measured by bits per character.

| Benchmark | HumanEval | Mbpp | DS-1000 | Average |
|---|---|---|---|---|
| Pearson $\rho$ | -0.903 | -0.916 | -0.946 | -0.937 |
| RMSE $e$ | 0.050 | 0.055 | 0.032 | 0.040 |

Table 7: The correlation for coding ability. Here, Pearson $\rho$ denotes the Pearson correlation coefficient, and RMSE $e$ represents the root mean square error.

**Mathematical Reasoning Ability:** The correlation between MMLU-Math and compression efficiency on the ArXiv-Math data is illustrated in Figure 9(c). The detailed evaluation results of mathematical reasoning ability are shown in Table 8. The breakdown of the correlation between benchmark scores and compression performance is presented in Table 9.

## C  Exploring the Impact of Random Sampling on Compression Evaluation

To investigate how random sampling affects results, we studied its impact on knowledge and commonsense ability, sampling 131M characters each time, as in our main experiment, for a total of three times from a pool of 1.3B characters for diversity. Results in Table 10 show minor variations between models, with an average standard deviation of 0.0021.

| Model | GSM8K | MATH | SAT-Math | MMLU-Math | Average | BPC on ArXiv-Math (↓) |
|---|---|---|---|---|---|---|
| | | | 6-7B Parameter Model | | | |
| Deepseek-llm-7b | 17.2 | 4.5 | 31.8 | 37.8 | 22.8 | 0.500 |
| Deepseek-math-7b | 59.2 | 28.0 | 60.0 | 50.5 | 49.4 | 0.420 |
| Falcon-7b | 3.9 | 1.0 | 29.5 | 34.7 | 17.3 | 0.541 |
| Llama-1-7b | 10.7 | 3.0 | 26.4 | 36.1 | 19.0 | 0.510 |
| Llama-2-7b | 13.7 | 3.5 | 28.6 | 39.5 | 21.3 | 0.500 |
| Llemma-7b | 31.7 | 14.9 | 52.7 | 39.6 | 34.7 | 0.431 |
| Mistral-7b | 33.4 | 11.8 | 48.6 | 46.1 | 35.0 | 0.443 |
| Qwen-7b | 42.8 | 13.4 | 43.6 | 41.3 | 35.3 | 0.483 |
| Yi-6b | 31.1 | 6.5 | 33.6 | 40.8 | 28.0 | 0.483 |
| | | | 13-15B Paramter Model | | | |
| Llama-1-13b | 13.6 | 4.1 | 28.6 | 39.0 | 21.3 | 0.487 |
| Llama-2-13b | 21.9 | 5.3 | 31.4 | 42.8 | 25.3 | 0.475 |
| Qwen-14b | 55.7 | 26.1 | 57.3 | 47.1 | 46.5 | 0.450 |
| | | | 30-40B Parameter Model | | | |
| Falcon-40b | 23.0 | 4.0 | 31.8 | 43.0 | 25.5 | 0.482 |
| Llama-1-30b | 32.1 | 6.6 | 36.4 | 44.9 | 30.0 | 0.456 |
| Llemma-34b | 46.2 | 20.3 | 54.5 | 48.0 | 42.3 | 0.416 |
| Yi-34b | 61.4 | 16.6 | 52.3 | 50.8 | 45.3 | 0.421 |
| | | | 50-72B Parameter Model | | | |
| Deepseek-llm-67b | 56.0 | 16.6 | 54.1 | 50.1 | 44.2 | 0.430 |
| Jamba-52b | 48.3 | 18.3 | 55.0 | 49.4 | 42.7 | 0.417 |
| Llama-1-65b | 42.5 | 10.0 | 41.8 | 45.3 | 34.9 | 0.441 |
| Llama-2-70b | 50.4 | 12.4 | 50.9 | 48.0 | 40.4 | 0.429 |
| Mixtral-8x7b | 53.4 | 23.7 | 64.1 | 54.4 | 48.9 | 0.394 |
| Qwen-72b | 72.6 | 36.2 | 71.4 | 52.3 | 58.1 | 0.415 |

Table 8: Results of mathematical reasoning ability. We report models' compression efficiency on the ArXiv-Math dataset measured by bits per character.

| Benchmark | GSM8K | MATH | SAT-Math | MMLU-Math | Average |
|---|---|---|---|---|---|
| Pearson $\rho$ | -0.922 | -0.899 | -0.933 | -0.916 | -0.953 |
| RMSE $e$ | 0.068 | 0.033 | 0.044 | 0.022 | 0.031 |

Table 9: The correlation for mathematical reasoning ability. Here, Pearson $\rho$ denotes the Pearson correlation coefficient, and RMSE $e$ represents the root mean square error.

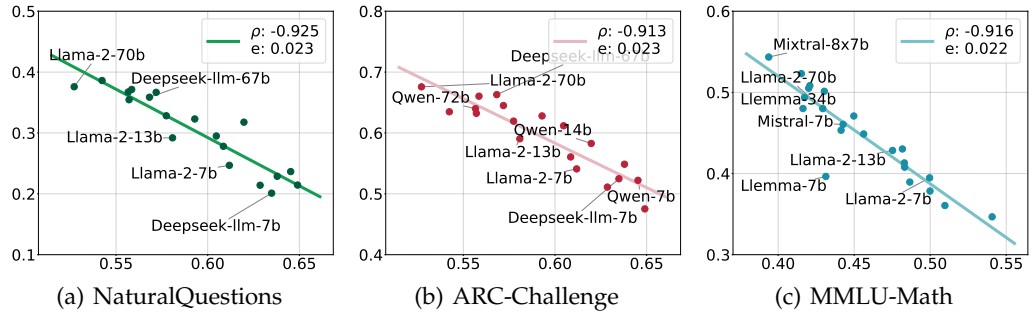

(a) NaturalQuestions     (b) ARC-Challenge     (c) MMLU-Math

Figure 9: The correlation between benchmark score and compression. Y-axis is the benchmark score while x-axis represents BPC. BPC in NaturalQuestions and ARC-Challenge are computed on Common Crawl data, while BPC in MMLU-Math is evaluated on the ArXiv-Math data.

# D   Exploring the Impact of Different Input Styles on MMLU Evaluation

Given the prominence of MMLU as a benchmark for LLMs, achieving exemplary scores is paramount. However, concerns arise if these practices rely on overfitting. For example, Bi et al. (2024) demonstrates that incorporating multiple choice questions (MCQs) into the

| Model | Average | Std |
|---|---|---|
| 6-7B Parameter Model | | |
| Deepseek-llm-7b | 0.635 | 0.002 |
| Falcon-7b | 0.650 | 0.002 |
| Llama-1-7b | 0.630 | 0.003 |
| Llama-2-7b | 0.613 | 0.002 |
| Mistral-7b | 0.606 | 0.002 |
| Qwen-7b | 0.646 | 0.002 |
| Yi-6b | 0.639 | 0.002 |
| 13-15B Paramter Model | | |
| Llama-1-13b | 0.610 | 0.003 |
| Llama-2-13b | 0.582 | 0.002 |
| Qwen-14b | 0.620 | 0.002 |
| 30-40B Parameter Model | | |
| Falcon-40b | 0.594 | 0.002 |
| Llama-1-30b | 0.579 | 0.002 |
| Yi-34b | 0.573 | 0.002 |
| 50-72B Parameter Model | | |
| Deepseek-llm-67b | 0.569 | 0.001 |
| Llama-1-65b | 0.559 | 0.003 |
| Llama-2-70b | 0.528 | 0.002 |
| Mixtral-8x7b | 0.560 | 0.002 |
| Qwen-72b | 0.558 | 0.002 |

Table 10: Results of Repeated Sampling on Common Crawl Dataset. This table presents the average bits per character obtained from three independent samplings. "Std" denotes the standard deviation of the results.

training stage can significantly improve the MMLU accuracy while achieving no performance gains for tasks not formatted as MCQs. Alzahrani et al. (2024) further observes that the performance of some models on the MMLU benchmark is highly sensitive to evaluation details, indicating potential overfitting. Therefore, in our main experiments, we intentionally opted for a less conventional evaluation style, namely the cloze style. The method involves presenting only the question without the options, and subsequently selecting the choice with the highest likelihood. It aligns with the standard way of evaluating ARC-Challenge (Clark et al., 2018), also an MCQ benchmark. By omitting multiple answer options, this approach deviates from the traditional MCQ format, thereby potentially reducing overfitting. Such a cloze-style format is just one of the many flavors of evaluating MMLU.[8] And we argue that this method provides a fairer assessment of MMLU's utility in gauging intelligence. For a thorough analysis of the impact of input styles, we employ the LM-Evaluation-Harness (Gao et al., 2023a) framework to evaluate the MMLU with the default style of input. The results, depicted in Figure 10, reveal a markedly weaker linear correlation between the scores and compression compared to the cloze-style input. Specifically, as demonstrated in Table 11, the scores for most models increase with the default input format, except Llama-1-7b and Falcon-7b. Notably, the Yi and Qwen series models exhibit substantial score changes, averaging nearly 20 within the series. These changes contribute to a significant deviation from the linear correlation, indicating potential overfitting on MCQ benchmarks.

## E  Better Alignment between MMLU and Textbooks

The MMLU benchmark offers a comprehensive evaluation across various domains and tasks, leveraging datasets sourced from real-world examinations and textbooks. As a result, textbooks from various educational levels closely align with the benchmark's distribution. We utilize a small collection of textbooks sourced from the internet as validation data. The results are shown in Figure 11. A stronger linear correlation is observed between the

---

[8]https://huggingface.co/blog/open-llm-leaderboard-mmlu

| Model | Default Style | Cloze Style | Δ |
|---|---|---|---|
| 6-7B Parameter Model | | | |
| Deepseek-llm-7b | 49.3 | 43.1 | 6.2 |
| Falcon-7b | 27.7 | 41.9 | -14.3 |
| Llama-1-7b | 35.0 | 43.1 | -8.1 |
| Llama-2-7b | 46.6 | 44.3 | 2.3 |
| Mistral-7b | 63.5 | 50.3 | 13.1 |
| Qwen-7b | 59.7 | 44.3 | 15.4 |
| Yi-6b | 64.0 | 45.8 | 18.2 |
| 13-15B Paramter Model | | | |
| Llama-1-13b | 47.1 | 46.1 | 1.0 |
| Llama-2-13b | 55.6 | 47.6 | 8.1 |
| Qwen-14b | 67.9 | 48.6 | 19.3 |
| 30-40B Parameter Model | | | |
| Falcon-40b | 57.1 | 48.9 | 8.2 |
| Llama-1-30b | 58.3 | 50.3 | 8.0 |
| Yi-34b | 76.4 | 55.0 | **21.5** |
| 50-72B Parameter Model | | | |
| Deepseek-llm-67b | 71.7 | 54.8 | 16.9 |
| Jamba-52b | 67.1 | 53.6 | 13.5 |
| Llama-1-65b | 63.6 | 51.4 | 12.3 |
| Llama-2-70b | 69.4 | 54.6 | 14.8 |
| Mixtral-8x7b | 71.7 | 56.5 | 15.2 |
| Qwen-72b | 77.4 | 54.5 | **22.9** |

Table 11: Comparison of different input style on MMLU. We report the average accuracy across all subjects. And we also report Δ as the difference of score between default style and cloze style.

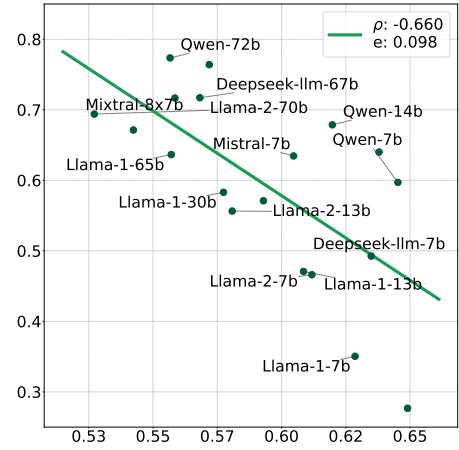

Figure 10: Results of MMLU with default style of input. The x-axis represents the models' compression efficiency on Common Crawl data, while the y-axis depicts the accuracy of MMLU.

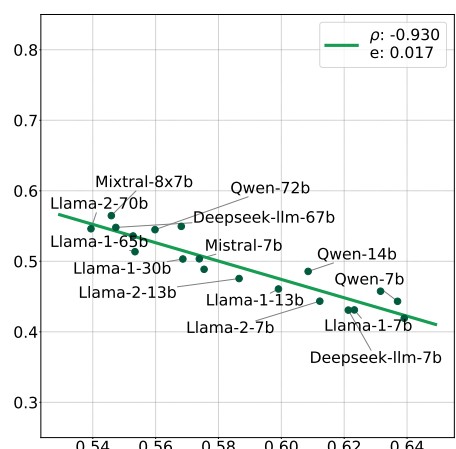

Figure 11: Correlation between MMLU accuracy and models' compression efficiency on textbook.

compression performance on the textbook data and the MMLU accuracy, characterized by a high Pearson correlation coefficient of -0.930.

