# OpenReview forum: "Compression Represents Intelligence Linearly"
_colmweb.org/COLM/2024/Conference — COLM_

### Official Review · Reviewer_Xsbe · 2024-05-10

**Rating:** 7
**Confidence:** 4
**Ethics Flag:** 1

**Summary:**

The paper examines the relationship between text compression efficiency and the intelligence of large language models. "Intelligence" is represented by performance on a selection downstream tasks such as commonsense reasoning and maths. Compression efficiency is measured as bits per character and derived from the conditional probabilities of a model on an unseen text corpora related to the benchmark tasks.
Experiments with a variety of models and benchmarks demonstrate an almost linear correlation, i.e. a model that compresses text better also performs better on benchmarks.
The authors propose the use of compression efficiency as a robust, unsupervised metric for evaluating LLMs, supporting the previous use of perplexity as a metric.

**Questions To Authors:**

* The authors did not directly compare text compression to alternative characteristics that could put the findings into more perspective. For example, is compression much more correlated to performance than, for instance, model size or training data size? This might already be implied but showing explicitly that other signals cannot predict performance/intelligence as well as compression would give more weight to the findings.
* The importance of the **compression** aspect isn't entirely to me - as the authors mention, there is a direct link between concepts such as cross-entropy loss, average bits for compression and perplexity. Could the title equally be 'perplexity represents intelligence linearly'? Some clarification around this would help.

**Reasons To Accept:**

* Interesting and relevant topic: understanding how the capabilities of LLMs relate to other signals is promising both for advancing the theory around language modelling and for practical implications.
* Strong experimental methodology and analysis; the authors also made good efforts to avoid data contamination and overfitting.
* The paper is very well-written and easy to follow.

**Reasons To Reject:**

* Lack of comparison of compression with alternative measures that could give similar correlations (see question below)
* Focus on the concept of compression (rather than 'level of surprisal' etc) not very clear (see question below)

---

> ### Author Rebuttal · Authors · 2024-05-31
>
> Thanks for your encouraging review! We address your comments below.
> ‎
> > The authors did not directly compare text compression to alternative characteristics that could put the findings into more perspective. For example, is compression much more correlated to performance than, for instance, model size or training data size?
>
> Thanks for the advice! We strongly believe that model size or training data size alone does not correlate well with performance across **different model families** — other factors such as model architectures and training data quality will influence the scaling law with model size or training data size as inputs alone. Although this may already be implied, as suggested by the reviewer, in the next revisions we will supplement such comparisons to explicitly show the comparison to give more weight to our findings.
>
> > The importance of the compression aspect isn't entirely to me - as the authors mention, there is a direct link between concepts such as cross-entropy loss, average bits for compression and perplexity. Could the title equally be 'perplexity represents intelligence linearly'? Some clarification around this would help.
>
> This is a good point. We emphasize compression in this paper due to the longstanding and recently revitalized discussion on the relation between compression and intelligence—a discussion that directly inspired our research. Indeed, there is a direct link between these concepts including losses, perplexity, and BPC. While there are various ways to present our experiments, the reviewer's suggested title, "Perplexity Represents Intelligence Linearly," is also valid. We chose to focus on compression because of its historical significance in discussions about intelligence, reflecting our original motivation and preference. However, presenting our findings from the perspectives of loss or perplexity is feasible as well.

---

> > ### Comment · Reviewer_Xsbe · 2024-06-05
> >
> > Thank you for addressing my points! I think focusing on the compression term makes sense - I would just make sure that it is clear in the paper how this is related to the already used practice of evaluating with perplexity, e.g. especially if there are any differences in implementation details that would affect the linearity.

---

> > > ### Author Response · Authors · 2024-06-07
> > >
> > > Thanks for the suggestion! The implementation details do matter a lot for the linearity. For example, as discussed in Section 3.2, if we do not unify the tokenization or context window size of different model families when computing BPC, the strong correlation will significantly deteriorate based on our initial observations. We will add some ablation results on this to make it clear.

---

### Official Review · Reviewer_Nicz · 2024-05-20

**Rating:** 5
**Confidence:** 4
**Ethics Flag:** 1

**Summary:**

This paper presents an observation that bits per character (BPC) is linearly aligned with the model's capability (average score) on multiple benchmarks. The BPC can be calculated via a variation of cross-entropy loss. Basically, if the model achieves better BPC (lower XE loss) on standalone evaluation sets, the model will perform better on the corresponding benchmarks. The author argues the BPC is a more reliable metrics than the standard benchmark.

**Questions To Authors:**

Please address the reasons to reject.

**Reasons To Accept:**

The observation that Bits per character is linearly aligned with the average benchmark performance on the corresponding categories is interesting, which I think might be the major reason to accept this paper.

**Reasons To Reject:**

Concretely speaking, this paper uses standalone (held-out) set to evaluate the BPC of the LLM. And BPC is a variation of the Cross Entropy loss. So the BPC calculation is basically doing an evaluation on held-out (validation) datasets (please correct me if I understand incorrectly.) If so, this is a standard ML practice. I might see limited novelty in this approach. If the author claims that the BPC of the training set is linearly aligned with the benchmark score on the held-out dataset, I think it will be more interesting. Unfortunately, as multiple LLM models are close-sourced / only released weights, it is impossible to validate the second (more interesting in my opinion) direction.

The author also claims the BPC is a reliable evaluation metric, which I agree, given the strong linearly relationship b/w the BPC and the benchmark score. However, as the average of the benchmark score is an established/well-accepted evaluation approach. I don't see the reason we need an additional one.

Further the author didn't claim / merely claim that BPC is better / more reliable than the benchmark score. However, I didn't see (might miss) the concrete evidence to establish the connection.

---

> ### Author Rebuttal · Authors · 2024-05-31
>
> Thanks for your helpful review! We address your comments below.
>
> > So the BPC calculation is basically doing an evaluation on held-out …… If so, this is a standard ML practice. I might see limited novelty in this approach.
>
> You are right. From a high-level view, this is indeed a standard ML practice. In LLMs, however, people typically compare losses either between different training steps within the same model, or among models within the same family (e.g., llama-7B, llama-13B). Prior to this study, there was no established norm for comparing losses **across different model families**  — for example, comparisons need to account for variations in tokenizers via the BPC metric and unify the context window size to ensure meaningful results. Additionally, it remains unclear whether comparing model losses across various model families — which differ in architecture, tokenizers, context window, and training data— is truly meaningful and has implications on downstream tasks. In this work, we establish a standard practice for making such comparisons, and provide empirical evidence that the relationship between compression and intelligence stands as a universal rule that is agnostic to the specific model and training configurations.
>
> > If the author claims that the BPC of the training set is linearly aligned with the benchmark score on the held-out dataset, I think it will be more interesting.
>
> While we cannot perform such experiments due to limited access to the training data, we don’t think the training losses of different models are directly comparable since the training losses are computed on different training data — this is exactly why we need to compute BPC on the same held-out corpus.
>
> > However, as the average of the benchmark score is an established/well-accepted evaluation approach. I don't see the reason we need an additional one
>
> As an unsupervised evaluation metric, BPC has practical advantages. Specifically, as discussed in the paper, data leakage and overoptimization are major issues for these benchmarks. For instance, using the MIN-K% PROB method for detection, we discover that the Qwen series have inflated scores on math benchmarks. In such cases, BPC serves as a more reliable metric. Also, BPC can easily support dynamic evaluation to update the compression corpora automatically to mitigate the leakage or overfitting issue, and it is easy to collect new domains of corpora. However, it is expensive to update or create new downstream benchmarks.

---

> ### Comment · Reviewer_Nicz · 2024-06-06
>
> Thanks for the rebuttal.
>
> For 1. Please correct me if I am wrong. From my perspective, "comparing losses across different model families" is basically the same as comparing somewhat 'metrics' on a held-out dataset across different model families. I think this is the common practice in multiple domain. e.g. comparing different image understanding architectures' generalization capability by comparing imagenet accuracy b/w VGGNet, AlexNet, and ResNet. Unfortunately, I don't think I am persuaded that this is considered as a new novelty.
>
> For 3. Just a quick check, IIUC, I think data leakage would also affect the BPC?

---

> > ### Author Response · Authors · 2024-06-07
> >
> > Thanks for the response.
> >
> > As in the previous rebuttal, we do agree with the reviewer that the concept of "comparing some metrics on a held-out dataset" is not new, yet we emphasize that the main point of this paper is not to propose a brand new evaluation approach, but to demonstrate the **linear correlation** between downstream performance and losses across different model families, which has not been revealed before and we don't think it was obvious that such a strong linear correlation exists across different model families. Additionally, while "comparing losses" is not something new, comparing losses across different model families requires to take care of several factors properly -- as we mentioned in the rebuttal --  such as accounting for variations in tokenizers via the BPC metric and unifying the context window size. These specific factors were often overlooked in previous research, but we have observed that failing to properly address these factors can completely disrupt the strong correlation with downstream results.
> >
> > For 3 on the data leakage discussion, you are right that data leakage will also influence BPC. However, here the difference is that the dataset for BPC can be automatically refreshed without significant human efforts -- just as in the paper, we simply run a pipeline script to obtain the corpora to test BPC. We can always refresh the corpora to the latest data easily through executing a program, which will mitigate the leakage issue through using the newest data. This is what we meant as dynamic eval since the eval data can be dynamically changed automatically. In contrast, most downstream task benchmarks are static and rarely updated, because expensive human efforts are required to create new datasets there.

---

### Official Review · Reviewer_hxZb · 2024-05-21

**Rating:** 5
**Confidence:** 4
**Ethics Flag:** 1

**Summary:**

The work is the first to document the linear correlation between compression and intelligence in LLMs.

**Reasons To Accept:**

1. The authors find the linear correlation between language model loss (bits per character) and downstream tasks.
2. The authors have a wide exploration of different LLM son three different domains.

**Reasons To Reject:**

1. The claimed linear correlation looks not surprising to me. Language model loss has always been an important metric to justify the quality of LLM. The paper is more on empirical study without proposing new idea,
2. More interesting experiments needed. It would be better if the authors could explore more domains. And it would be better if the authors could find something across domains, e.g. how Github data correlates to NQ, MMLU, etc.
3. Figures 1,2,3,4 look duplicated. Figures 2,3,4 can be merged if some information has been shown in Figure 1.

---

> ### Author Rebuttal · Authors · 2024-05-31
>
> Thanks for your time and helpful review! We respond to your comments below.
>
> > The claimed linear correlation looks not surprising to me
>
> LM losses have indeed been an important metric, yet its relation to downstream task performance across different LLM families remains unknown. We highlight two novel findings:
> 1. In LLMs, typically, people only compare losses either between different training steps within the same model, or among models within the same family (e.g., llama-7B, 13B). These models share most design and training configurations. However, it is unclear whether comparing losses of LLMs from **different model families** has implications for downstream task results. Our work is the first to examine this. Note that these LLMs differ in tokenizer, architecture, context window size, and training data. We don’t think that it is immediately apparent that the correlation universally holds and is **agnostic** to all these factors, a fact only revealed through our experiments.
> 2. The correlation between losses and downstream task scores across different LLM families, if it exists, can take many forms. Our research reveals that it is linear.
>
> > The paper is more on empirical study without proposing new idea.
>
> Right, the paper is more on empirical study, which we believe reveals novel insights. We don’t think that emphasizing empirical study is a weakness of a paper.
>
> > More interesting experiments needed … it would be better if the authors could find something across domains, e.g. how Github data correlates to NQ, MMLU, etc.
>
> First, as we already showed in Figure 6, the BPC on Github data is not well correlated with the average score on the knowledge domain that includes MMLU and NQ.
>
> To add more experiments as requested, we explore cross-ability tasks where multiple abilities are required. We consider solving GSM8K questions through coding, which requires both math and coding abilities. We compute BPC on three corpora to examine their correlation with GSM8K scores: (1) on arxiv data; (2) on Github data; and (3) on the mix of arxiv and Github data. Results are shown below — as expected, the mix of arxiv and Github data demonstrates the strongest correlation, while either arxiv or Github data alone does not correlate well, underscoring that different corpora reflect diverse abilities.
>
> | Corpora | Pearson $\rho$  | RMSE $e$ |
> | --- | --- | --- |
> | Github alone | -0.742 | 0.137 |
> | ArXiv alone | -0.787 | 0.126 |
> | Github+ArXiv | -0.940 | 0.070 |

---

### Official Review · Reviewer_pVVB · 2024-05-21

**Rating:** 7
**Confidence:** 4
**Ethics Flag:** 1

**Summary:**

The study explores the correlation between data compression and intelligence in the context of LLMs, positing that better compression efficiency correlates with higher intelligence. Experiments on 30 public LLMs across 12 benchmarks showed a near-linear correlation between compression efficiency and intelligence.

**Questions To Authors:**

1. Considering the rapid pace of innovation in the field of LLMs, is there confidence that the findings would remain applicable to novel architectures such as RWKV, Mamba, and KAN?
2. Is there an expectation that the correlation between compression efficiency and intelligence persists across different languages or modalities?

**Reasons To Accept:**

1. This work provides substantial empirical evidence supporting the hypothesis that superior compression indicates greater intelligence.
2. The study employs a robust methodology, using a large and diverse set of LLMs and benchmarks to somewhat ensure the generalizability of the findings.
3. The findings suggest compression efficiency as a practical and unsupervised metric for evaluating LLMs, which can be beneficial for future research and development.

**Reasons To Reject:**

1. The definition of intelligence in the paper is limited to performance on specific benchmarks, which might not capture all dimensions of intelligence.
2. As mentioned in the paper, the study focuses on short- to medium-context scenarios, which may not fully represent the models' capabilities in long-context tasks.
3. While the paper encompasses a variety of models, it remains uncertain whether its conclusions are applicable to novel or significantly different architectures [1,2,3] not addressed.

[1] Peng, Bo, et al. "Rwkv: Reinventing rnns for the transformer era." arXiv preprint arXiv:2305.13048 (2023).
[2] Gu, Albert, and Tri Dao. "Mamba: Linear-time sequence modeling with selective state spaces." arXiv preprint arXiv:2312.00752 (2023).
[3] Liu, Ziming, et al. "Kan: Kolmogorov-arnold networks." arXiv preprint arXiv:2404.19756 (2024).

---

> ### Author Rebuttal · Authors · 2024-05-31
>
> Thanks for your time and encouraging review! We respond to your comments below.
>
>
> > Considering the rapid pace of innovation in the field of LLMs, is there confidence that the findings would remain applicable to novel architectures such as RWKV, Mamba, and KAN?
>
> We hypothesize that the findings are agnostic to model architectures, while we did not test non-transformer architectures since there were few non-transformer pretrained models with strong and already-emergent abilities. To verify the findings on non-transformer archs, however, we conduct the experiment on the Jamba-52B model [1], which is based on a hybrid Transformer-Mamba MOE architecture, being the strongest open-source non-transformer pretrained model as far as we know.  The results for Jamba model in the main figure are shown in [this link](https://drive.google.com/file/d/15z6dpF_hGwP5l0w8lXVprFZxp23mm9k9/view?usp=sharing), while we report its BPC and benchmark scores below (according to the COLM rebuttal guideline, we think this is the necessary case to link such an external correlation figure that is easy to read, since the numbers below alone do not reveal much helpful information).  Despite having significantly different architectures, our findings can still be applicable to the Jamba model.
>
>
>
> | Ability                   |  BPC  | Average Benchmark Score |
> | ------------------------- | :---: | :---------------------: |
> | Knowledge and Commonsense | 0.542 |          0.624          |
> | Coding                    | 0.244 |          0.351          |
> | Mathematical Reasoning    | 0.417 |          0.413          |
> | Overall                   | 0.401 |          0.463          |
>
> > Is there an expectation that the correlation between compression efficiency and intelligence persists across different languages or modalities?
>
> This is a good point. We do expect the correlation between compression efficiency and intelligence persists across different languages or modalities, because the long-standing discussion on their relationship has been based on data in general rather than language only. However, in this paper we focus on the single-language setting and leave further exploration of different languages and modalities as future work.
>
> [1] Lieber, Opher, et al. "Jamba: A hybrid transformer-mamba language model." *arXiv preprint arXiv:2403.19887* (2024).

---

### Official Review · Reviewer_4KvH · 2024-05-22

**Rating:** 9
**Confidence:** 4
**Ethics Flag:** 1

**Summary:**

This work analyzes the relationship between the ability to compress a text corpora and the ability to solve downstream tasks associated with such corpora. This approximates the relationship between compression (treating an LLM as a compressor) and intelligence (as an ability to solve a task well). Across a wide range of models and scales, the overall message that intelligence and compression have a strong linear relationship is both novel and surprising. This work does an excellent job of dissecting their claims, providing multiple modes of analysis and giving broad context and supporting material.

**Reasons To Accept:**

- The linear relationship is novel, surprising and insightful. This will be extremely discussion worthy at COLM. Other reviewers have asked for impossible comparisons (e.g., to KAN, a paper which dropped literally 3 weeks ago), or to be wowed by some super unsurprising result. I respectfully disagree with all these reviews, and consider this work a solid piece of science which is important to the community. If I could hit the Best Paper recc button, I would.

- The authors elagantly propose multiple modes of analysis which future practitioners can use. The usage of correlation, bit coding and statistical analyses in this paper are exemplars for the community which others should follow.

- Quantifying two very abstract concepts into quantitative measures is useful for further discussion in the community when we discuss what an LM "knows" and "does".

- A wide range of models are explored to make results more convincing and reliable

- The domains of code, math, and commonsense are sensible and provide reliable bases to easily interpret the findings. The results are communicated excellently.

**Reasons To Reject:**

- Nothing major. The authors could have explored very small model sizes to see if these findings extend to GPT2-base or BERT size models. I do not agree with the demands to explore multiple architectures, as many now consider architecture to be very much secondary to data. This paper shows the importance of data, and should be considered much more valuable than evaluation on many architectures.

---

> ### Author Rebuttal · Authors · 2024-05-31
>
> Thank you for your time and encouraging review!

---

> > ### Comment · Reviewer_4KvH · 2024-06-07
> > **Response Required**
> >
> > Thank you for your response.

---

### Decision · Program_Chairs · 2024-07-10

**Decision:**

Accept

**Comment:**

This paper studies the correlation between data compression and intelligence in the context of LLMs, positing that better compression efficiency correlates with higher intelligence. After author rebuttal, it still received mixed scores of 55779.

On one hand, three of the reviewers are positive about the paper, commenting that (1) the finding in the paper is both novel, surprising, and insightful, (2) the analysis and experiments are well conducted, and (3) the paper is well written. On the other hand, the other two reviewers commented that the claimed linear correlation looks not surprising to me, and more experiments are needed.

The authors did a nice job of rebuttal, and added additional results of Jamba. Overall, the studied research topic in this paper is quite interesting, and will probably raise meaningful discussions in the conference and the community. The AC thinks that the merits outweigh the flaws, and thus, would like to recommend acceptance of the paper.